# Characterization of Extremely Fresh Biomass Burning Aerosol by Means of Lidar Observations

Benedetto De Rosa [1,*], Francesco Amato [1], Aldo Amodeo [1], Giuseppe D'Amico [1], Claudio Dema [1], Alfredo Falconieri [1], Aldo Giunta [1], Pilar Gumà-Claramunt [1], Anna Kampouri [2,3], Stavros Solomos [2,4], Michail Mytilinaios [1], Nikolaos Papagiannopoulos [1], Donato Summa [1], Igor Veselovskii [5] and Lucia Mona [1]

1    Consiglio Nazionale delle Ricerche—Istituto di Metodologie per l'Analisi Ambientale CNR-IMAA, 85050 Potenza, Italy
2    IAASARS, National Observatory of Athens, 10560 Athens, Greece
3    Department of Meteorology and Climatology, School of Geology, Aristotle University of Thessaloniki, 54124 Thessaloniki, Greece
4    Research Centre for Atmospheric Physics and Climatology, Academy of Athens, 10680 Athens, Greece
5    Prokhorov General Physics Institute, 142190 Moscow, Russia
*    Correspondence: benedetto.derosa@imaa.cnr.it

**Abstract:** In this paper, characterization of the optical and microphysical properties of extremely fresh biomass burning aerosol is presented. This work aims to characterize, for the first time to our knowledge, freshly formed smoke particles observed only a few minutes after they were emitted from a nearby forest fire. The smoke particles were detected by combining passive (sun-photometer) and active (Raman lidar) techniques. On 14 August 2021, an EARLINET (European Aerosol Research Lidar Network) multi-wavelength Raman lidar and a co-located AERONET sun-photometer in Potenza, South Italy, observed an extremely fresh smoke plume. The lidar measurements, carried out from 22:27 to 02:16 UTC, revealed a thick biomass burning layer below 2.7 km. The particle depolarization ratio at 532 nm was 0.025, and lidar ratios at 355 and 532 nm were, respectively, 40 and 38 sr. The mean value of the Ångström exponent was 1.5. The derived size distribution was bimodal with a peak at 0.13 μm, an effective radius mean value of 0.15 μm, and a single scattering albedo of 0.96 at all wavelengths. The real part of the refractive index was 1.58 and the imaginary was 0.006. The AERONET measurements at 5:34 UTC confirmed the lidar measurements.

**Keywords:** fresh biomass burning; lidar Raman; sun photometer

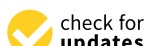



## 1. Introduction

Summer 2021 was characterized by exceptionally warm and dry climate conditions, especially in the Mediterranean, which caused a high number of forest fires [1]. The biomass-burning aerosols influence the radiation budget and the regional climate [2]. In the coming years, these events are bound to increase. Indeed, the Mediterranean Basin is one of the four most significantly altered hotspots on Earth [3], and climate models predict drier and hotter summers [4]. Biomass burning (BB) episodes measured at 14 stations of the European Aerosol Research Lidar Network (EARLINET; www.earlinet.org) over 2008–2017 [5] reveal that in Southeast Europe (Potenza, Athens, Bucharest, Sofia, Thessaloniki lidar stations), the origin of the smoke was predominantly local (only 3% originated in North America). These fires, even if small in size, are very high in number, and their climatic impacts cannot be neglected. Despite their importance, the literature on very fresh biomass burning (<24 h) is very poor. Fresh biomass burning aerosols consist of elemental carbon (EC) and organic carbon (OC). EC, usually associated with the black carbon component, primarily absorbs solar radiation. Organic carbon primarily scatters solar radiation, but is still poorly characterized, both chemically and optically [6]. Small fires are often extinguished within a few hours; therefore, they do not reach high

temperatures, and the dominant fraction consists of OC. Current climate models likely overestimate the contribution of these fires. This study shows how in the case of local fires, the absorption of electromagnetic radiation is lower compared to the scattering component.

A further parameter to be considered in the characterization of biomass burning particles is the vegetation type, which affects the optical properties and the single scattering albedo (SSA) of the smoke particles. Typically, smoke originating in North America scatters more and absorbs less than fire in Brazil because of the smaller size of the biomass burning particles in Brazil [7]. Furthermore, a large component of the total fuel consumed during the fire in North America is related to the smoldering of the forest subsurface. The smoke plumes produced are poor in black carbon and show low absorption. Usually, the optical properties are analyzed by ground-based radiometers and satellites [8]. However, these techniques cannot characterize accurately the high variability of the vertical structure of smoke aerosol. To analyze isolated layers, aircraft sampling can be used, but regular measurements are too expensive. Raman lidars are characterized by high temporal and vertical resolutions and have demonstrated a strong capability to study optical properties and the vertical structure of forest fire smoke [9,10]. In the past, important information about smoke particle parameters, such as lidar ratio, particle depolarization ratio, and Ångström exponent, has been provided by lidar measurements performed within EARLINET [9,11,12]. However, there is a wide range of optical and microphysical parameters to take into account for a complete characterization of biomass burning aerosol [13]. Generally, small and spherical smoke particles are characterized by low depolarization ratio, high Ångström exponents, and large lidar ratios [9]. In particular, the depolarization ratio is about 0.1, the Ångström exponents are approximately 1.5 at all wavelengths, and lidar ratios are between 60 and 90 sr [14]. Lidar ratio values for fresh smoke are generally lower compared to aged smoke [9].

Lidar measurements allow for microphysical aerosol parameters to be obtained through inversion algorithms. High-quality optical data along with depolarization information are required [15]. In these conditions, inversion algorithms allow for the retrieval of the complex refractive index, which is a crucial parameter to understand the optical properties and chemical composition of the particles. Other microphysical properties obtained are the size distribution; the number, surface, and volume concentration; the effective radius; and the single scattering albedo.

This paper reports measurements carried out with the Raman lidar system MUSA (Multiwavelength System for Aerosol) deployed at the CNR-IMAA Atmospheric Observatory (CIAO) in Potenza (Italy). In addition to lidar, AERONET (Aerosol Robotic Network) sun photometer measurements were used. Synergy with co-located lidar measurements is very important. Several studies [16,17] have shown that measurements by sun-photometer can assist the optical properties retrieved by lidar. Indeed, they minimize the uncertainty on the a priori assumptions needed for lidar inversion. This paper presents for the first time, to our knowledge, a characterization of vertical resolved optical and microphysical properties for an extremely fresh biomass burning plume. This provides insight into small and short-lived forest fires. These events are recurrent and numerous in the Mediterranean Basin. The paper is organized as follows: Section 2 describes the instruments used, and Section 3 presents the forest fire event that occurred on 14 August 2021, along with a detailed analysis of the aerosol optical and microphysical properties obtained by lidar and sun photometer measurements. Conclusions are reported in Section 4.

## 2. Experimental Set-Up

The measurements were collected at the CNR-IMAA Atmospheric Observatory (CIAO) [18]. CIAO is located in the middle of the Mediterranean Basin, on the Apennine mountains (40.60 N, 15.72 E, 760 m asl), and less than 150 km from the Tyrrhenian and the Ionian Sea. The observatory operates in a typically mountainous climate, strongly influenced by the Mediterranean atmospheric circulation and by the local orography. Summers are generally dry and hot, and winters are cold. The location allows for the study of different aerosol types in different meteo-

rological conditions. CIAO is one of the largely equipped atmospheric observatories in southern Europe, and it is a key site for ACTRIS, the European Research Infrastructure for aerosol, clouds, and trace gases investigation (www.actris.eu). CNR-IMAA is leading the node of the ACTRIS data Center for aerosol remote sensing (ARS) and is part of the ACTRIS center for aerosol remote sensing (CARS). Furthermore, CIAO is an ACTRIS National Facility for aerosol remote sensing [19], i.e., co-location of Raman lidar with a photometer, operating since 2000 and 2004, respectively. In the following, the two instruments are briefly described.

### 2.1. Multiwavelength Raman Lidar

MUSA is one of the EARLINET reference lidar systems that comprises the basis of the aerosol remote sensing component of ACTRIS. MUSA is developed around a Nd: YAG laser source. Laser pulses at 1064, 532, and 355 nm are simultaneously transmitted into the atmosphere at a zenith angle. The telescope is in Cassegrain configuration, with a primary mirror diameter of 0.3 m. The system is able to detect the backscattered Raman radiation by atmospheric $N_2$ molecules at 607 and 387 nm and elastic backscattered radiation at 1064 and 355 nm, while for 532 nm, cross and parallel components are detected individually, being separated through a beam splitter polarizing cube. The detection is performed through photomultiplier tubes, except for 1064 nm, which is detected with an avalanche photodiode (APD). The backscattered radiation at all the wavelengths is acquired both in analog and photon counting mode, except for 1064 nm, which is acquired only in analog mode. The calibration of the 532 nm depolarization channel is performed using the Δ90 method [20]. MUSA performs routine quality-assurance tests in the frame of the EARLINET/ACTRIS quality-assurance program to harmonize the lidar measurements, maintain high-quality standards, and improve the lidar data evaluation [21,22]. The temporal resolution of the raw profiles is 1 min, and the vertical resolution is 3.75 m. MUSA is not an automatic system; therefore, the measurements are not continuous and usually extend over a time window of three hours. This configuration allows for the retrieval of aerosol extensive and intensive properties. The extensive properties are the aerosol extinction coefficients at 355 and 532 nm and the aerosol backscatter coefficients at 355, 532, and 1064 nm. The intensive properties are the lidar ratio at 355 and 532 nm, the particle linear depolarization ratio at 532 nm, and the Ångström exponents using the pairs 355 nm and 532 nm, 532 nm and 1064 nm, and 355 nm and 1064 nm [23]. The intensive properties are independent of the aerosol load, contrary to the extensive properties, and depend only on the nature of the specific aerosol type.

The backscatter coefficients at 1064, 532, and 355 nm and the extinction coefficients at 355 and 532 can be inverted into particle microphysical properties using regularization techniques [15,23–25]. These techniques are based on the so-called Fredholm integral equations of the first kind describing the relation between optical data and the corresponding microphysical aerosol properties. Five integral equations are solved using Tikhonov's regularization method with appropriate constraints [26]. As the inverse problem is strongly underdetermined, a large number of a priori assumptions in the retrieval is necessary. Therefore, the parameters that are considered in the inversion cover a limited range of values. A detailed description of the approach can be found in the literature [15,27,28]. The technique is used to estimate microphysical outputs within data inside some representative height intervals where 3 backscatter and 2 extinction coefficients are available. After the particle volume size distributions are obtained, it is possible to retrieve the volume and surface area concentration and particle effective radius. The complex refractive index is also obtained from the measurements. The particle size distributions and complex refractive indices are subsequently used to calculate the particle single-scattering albedo.

### 2.2. AERONET Sun Photometer

The CIMEL CE318 sun photometer [29] provides measurements of aerosol columnar properties, and it is operational at CIAO in the frame of AERONET [29], the main network of sun photometer measurements on a global scale. It is a fully automatic sun-and-sky

radiometer measuring sky radiance and direct solar irradiance at different wavelengths. To remove the cloud contamination, a cloud masking procedure is applied. The final products are columnar measurements of water vapor; aerosol optical depth (AOD) at 340, 380, 440, 500, 675, 870, 1020, and 1640 nm; single scattering albedo (SSA), size distribution, refractive index, absorption optical depth, extinction optical depth, and asymmetry factor. The measured radiances are automatically sent to the NASA-GSFC where they are processed according to the AERONET data analysis. The co-location of passive and active remote sensing techniques such as Raman lidar and sun-photometer provides the opportunity to exploit an excellent synergy for the characterization of atmospheric aerosol, and it is a requirement for the ACTRIS aerosol remote sensing facilities.

### 3. Results: Forest Fire Event on 14 August 2021

Biomass burning aerosol from an oak forest fire was monitored by the neighboring CIAO, which is located only 1 km away from the emitting fire. The fire started at 16:00 UTC and was totally under control by the local fire brigade around 23:00 UTC. The extent of the burnt area and the locations of both the forest fire and CIAO can be seen in Figure 1. Data from the MSI (MultiSpectral Instrument) onboard the Copernicus Sentinel-2 were used to estimate the burnt scar. Specifically, we used MSI Level 2 data in the NIR and SWIR for a pre-fire and post-fire satellite overpass (i.e., 10/08/2021 at 09:50 UTC and 15/08/2021 at 09:50 UTC) following the methodology developed by Keeley [30]. The calculated index refers to different severity levels, and the burnt area covered an area of 0.061 km$^2$.

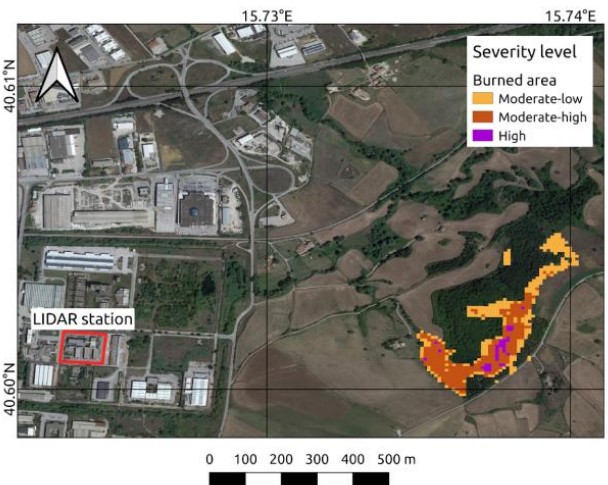

**Figure 1.** The fire severity map, estimated with MSI/Sentinel-2 data, indicates that the burnt oak forest fire occurred on 14 August 2021 near the ACTRIS/EARLINET station (red box).

The BB smoke dispersion was simulated using the Lagrangian particle dispersion model FLEXPART-WRF [31,32] in the forward mode. FLEXPART-WRF simulations were driven by hourly meteorological fields from the Advanced Research WRF (ARW) model version 4 [33]. The WRF-ARW spatial setup was at a 9 × 9 km resolution domain with 600 × 370 grid points and 31 vertical levels. Simulations were initiated at 00:00 UTC on 13 August 2021 and extended up to 00:00 UTC on 15 August 2021. The initial and boundary conditions for the offline coupled FLEXPART-WRF runs were taken from the National Centers of Environmental Prediction (NCEP) Global Forecast System (GFS) with 0.5° × 0.5° resolution. Sea Surface Temperature (SST) analysis data were provided by the Copernicus Marine Environment Monitoring Service (CMEMS) at a resolution of 1/12°. The use of 1-hourly WRF meteorological fields at a 9 × 9 km spatial resolution allows for a more detailed representation of the smoke plume dispersion. For the FLEXPART simulations, a total of 10,000 particles was released for each fire hot spot, and two modeling domains (Figure 2) were used for the simulations of the current study. Both domains in

FLEXPART-WRF were set up with 32 vertical levels (250–8000 m by a 250 m step), and at a horizontal resolution of $3° \times 3°$ in the outer domain and $0.1° \times 0.1°$ in the inner domain. The tracer particles were assumed to be smoke aerosol (total particulate matter—TPM). Dry and wet removal processes were also enabled for these particles.

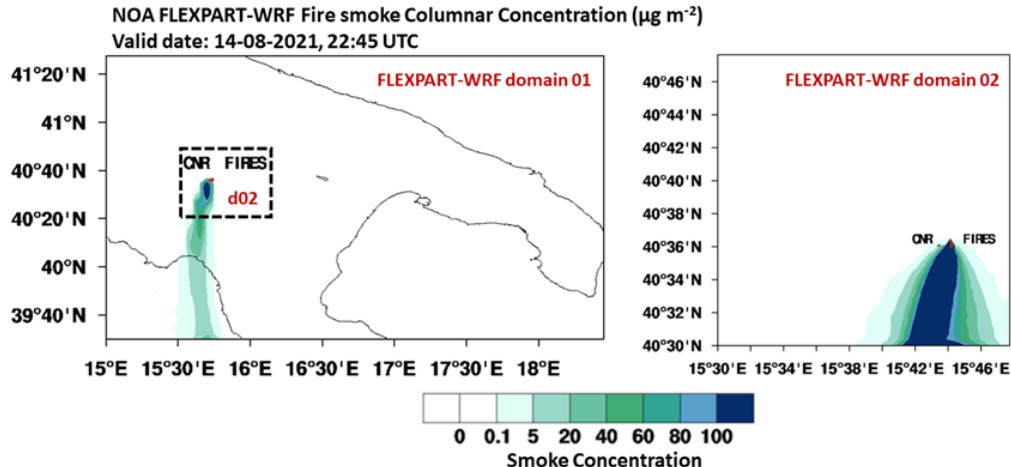

**Figure 2.** Simulated FLEXPART––WRF columnar concentration ($\mu g\ m^{-2}$) in two modeling domains for the smoke episode on 14 August 2021, 22:45 UTC: (**left panel**) $3° \times 3°$ (300 km), and (**right panel**) $0.1° \times 0.1°$ (11 km) horizontal resolution.

The dispersion simulation started on 14 August 2021, 16:00 UTC, when the fire near CIAO started, and ended at about 23:00 UTC. The simulated average columnar concentration of smoke TPM was spread to west–southwest directions over CIAO (Figure 3, left panel). Moreover, the near-surface concentration of smoke TPM in the inner domain (Figure 3, right panel) had maximum values of 103 $\mu g\ m^{-3}$, while over land and close to 8 km in height, the modeled columnar concentration had maximum values up to 500 $\mu g\ m^{-2}$, meaning that the highest TPM concentrations were found at higher levels than the surface.

The lidar measurements began at 22:27 UTC, 6 h after the fire started. The time–height evolution of the attenuated backscatter coefficient at 1064 nm (Figure 4) showed the presence of an intense aerosol layer since the beginning of the time series when the fire was still active. After 23:30 UTC, the intensity decreased considerably; however, it was possible to observe a residual layer under 2 km until the end of the measurement period. This could be related to the fact that in the case of small fires, the pyro-convection is not reached, and the plume remains confined inside the planetary boundary layer (PBL). In general, the height of the plume influences the transport and residence time in the atmosphere of the smoke. Within the PBL, winds have lower intensity, and the smoke remains well mixed near the emission source. Furthermore, the plume has a short lifetime in PBL due to deposition [34]. For this reason, after 00:52 UTC, the map showed a very intense layer at the surface level. The relative humidity for 14 and 15 August over Potenza as provided by the ECMWF forecast (https://hdl.handle.net/21.12132/1.65370c27fd0c437b; accessed on 26 September 2022) showed low values of around 25% in the PBL. This low humidity implied no hygroscopic growth for the smoke layers, but eventually a growth due to coagulation, which finally led to dry deposition.

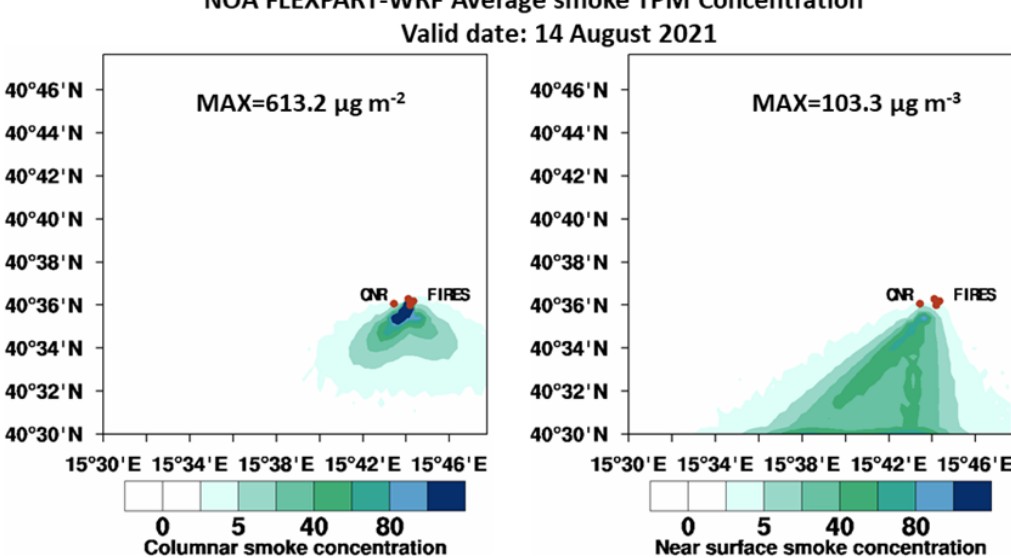

**Figure 3.** Simulated FLEXPARTò−WRF average smoke TPM concentration (**left**) columnar concentration ($\mu$g m$^{-2}$) and (**right**) smoke concentration ($\mu$g m$^{-3}$) at the first model layer on 14 August 2021.

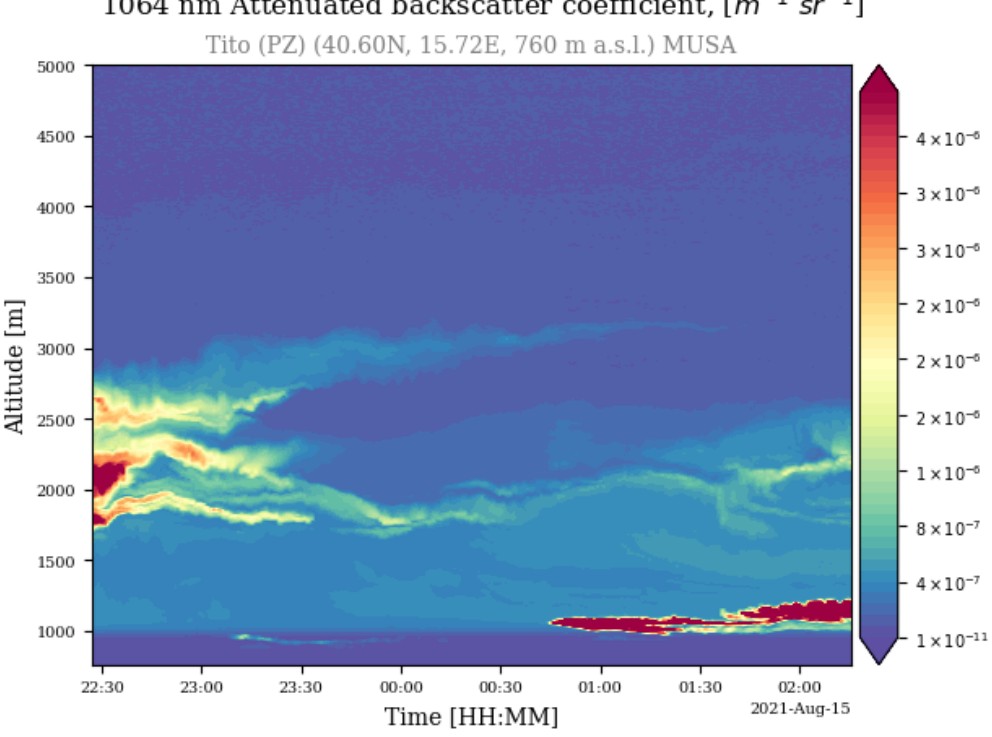

**Figure 4.** Temporal evolution of the attenuated backscatter coefficient at 1064 nm obtained with the MUSA lidar system on 14 August 2021 from 22:27 to 02:16 UTC.

### 3.1. Lidar Measurements

Two different periods with different characteristics were selected: the first, from 22:27 to 23:19 UTC when the fire was still active, and the second from 00:46 to 02:16 UTC when the fire was smoldering. The data were processed with the Single Calculus Chain (SCC), the centralized processing tool of EARLINET/ACTRIS [35].

From 22:27 to 23:19 UTC, the attenuated backscatter coefficient at 1064 (Figure 4) indicated the presence of an intense aerosol layer between 1.7 and 2.7 km. Figure 5 reports the

corresponding vertical profiles of the particle backscatter coefficient (β); particle extinction coefficient (α); lidar ratio (LR); Ångström exponents (AEβ) at 355–532, 355–1064, 532–1064 nm; particle linear depolarization ratio (PLDR); and volume linear depolarization ratio (VLDR) at 532 nm. The β and α profiles showed high values, while low PLDR (less than 5%) and VLDR (less than 4%) indicated the presence of mostly spherical particles [36,37]. The AOD at 532 nm obtained by integrating the extinction coefficient over the biomass burning layer was 0.153, whereas the total AOD was 0.261 (about 58% of the total AOD was due to the smoke layer). For 2.1–2.7 km, the mean values of LR at 355 and 532 nm were, respectively, 40.0 ± 2.2 and 38.0 ± 6.1 sr and were associated with low particle absorption. The values were similar to those found in the literature for fires where the black carbon component was minimal [22]. The values of AEβ were 1.5 ± 0.2 at all wavelengths, typical for moderately small particles in the accumulation mode, and were similar to the values associated with urban and industrial smoke [38].

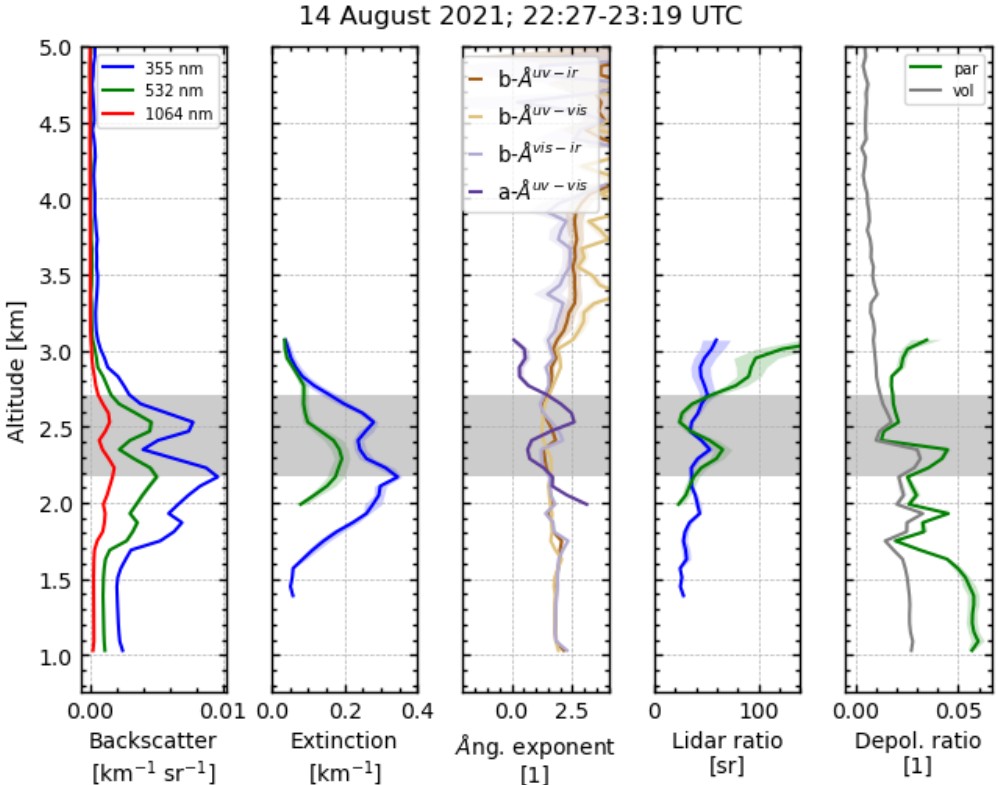

**Figure 5.** Vertical profiles of optical parameters measured in Potenza on 14 August 2021 from 22:27 to 23:19 UTC with the multi-wavelength Raman lidar MUSA. The grey rectangle shows the altitude from 2170 to 2710 m asl used for obtaining the microphysical particle properties through the inversion method.

Table 1 lists the optical and microphysical properties of eight biomass burning events of young smoke (<2 days) found in the literature; the authors in [22] reported a biomass burning event as fresh as the event studied in this paper.

The values of LR were generally higher than those measured by MUSA. However, it is interesting to note that the fire in the Amazon had values of LR355 and LR532 of 43 sr and 41 sr, respectively, which were very similar to those measured by MUSA. The reason is not clear for these unusually low values that correspond to extremely fresh biomass burning aerosol. Typically, LR values from 30 to 60 sr along with high Ångström exponents indicate weakly absorbing smoke particles [22]. It must be considered that the black carbon content can vary from 2% to 30% [7,39,40]. These concentration differences produce highly variable measurements of optical and microphysical parameters for biomass burning. Furthermore, Müller et al. [41] found variable lidar ratios ranging from 30 sr to 90 sr for Canadian and

Siberian Forest fires. The ratio of lidar ratio (RLR), which in our case is the ratio of LR532 to LR355, yielded values lower than 1. Many studies have shown that values of RLR < 1 are typical of fresh biomass burning aerosol, whereas values of RLR > 1 are indicative of aged biomass burning particles [12,23,41,42].

The microphysical properties of the abovementioned layer were retrieved through the lidar inversion method [15,43] using the $3\beta + 2\alpha$ optical profiles. In the time interval from 22:27 to 23:19 UTC, a range of altitudes from 2170 to 2710 m asl was chosen. The retrieved concentration was quite high with values of number concentration (N), surface concentration (S), and volume concentration (V) of 2300 $cm^{-3}$, 410 $\mu m^2\ cm^{-3}$, and 21 $\mu m^3\ cm^{-3}$, respectively. The mean effective radius was 0.15 $\mu m$, similar to previous studies. Reid and Honns [7] reported values between 0.12 $\mu m$ (fresh smoke) and 0.21 $\mu m$ (aged smoke). More recently, values from 0.19 to 0.44 $\mu m$ were measured for aged biomass burning aerosol layers [12] while for fresh smoke (1–3 days), values from 0.13 to 0.15 $\mu m$ were obtained [23]. However, none of the cases found in the literature refers to few-minute fresh smoke layers like those observed in this study. In Table 1 we can see how the fires in the Amazon and in Portugal were characterized by effective radii very similar to those reported here (0.13 and 0.15 $\mu m$, respectively). The retrieved refractive index (m) was 1.58–0.006, and the single scattering albedo (SSA) was 0.969 at 355 nm, 0.967 at 532 nm, and 0.953 at 1064 nm. These values are indicative of low absorption and an almost absent black carbon component. The values of refractive index and SSA available in the literature are quite variable. Some studies show good agreement with those presented in this study [44,45], while for other studies, the scattering component was significantly lower [41,46]. As already mentioned, low black carbon values are often associated with smoldering undergrowth [40]. The best agreement in terms of refractive index and SSA is found for the fire that occurred in Portugal, while, unfortunately, no comparison is possible for Baars et al. [22].

Figure 6 reports the aerosol optical parameters corresponding to the time window 00:46–02:16 UTC when a dense smoke layer was observed near the surface. At this stage, due to the end of the fire and the dry deposition, the concentration of the aerosol at higher altitudes decreased significantly. Nevertheless, a residual layer was still visible in the PBL. Between 1.4 and 2.2 km asl, the values of LR at 355 and 532 nm were $40.0 \pm 1.6$ and $50.0 \pm 3.5$ sr, respectively, and the RLR was greater than 1. The value of AE$\beta$ for the pair 355–532 nm was $1.5 \pm 0.3$, while for the 355–1064 nm and 532–1064 nm pairs it was $1.4 \pm 0.3$. Consequently, we could deduce that the particles underwent a slight increase in their size.

Furthermore, the volume size distribution, as reported in Figure 7, showed the presence of a bimodal distribution; therefore, the larger particles, even if present in lower concentrations, produced an increase in the overall scattering capacity. The first distribution peak was located at 0.13 $\mu m$ with values greater than 20 $\mu m^3\ cm^{-3}$, while the second was in the accumulation mode at 0.5 $\mu m$ with values of approximately 5 $\mu m^3\ cm^{-3}$. This bimodal distribution was not related to the aging process; indeed, the vegetation types and the ratio between flaming and smoldering produce a different kind of particle [47,48]. These differences were maintained for a limited period, as mixing and homogenization of the smoke aerosol happened in a short time. From 00:46 to 02:16 UTC, the altitude between 1510 and 2050 m was selected for the characterization of the microphysical properties. The values of N, S, and V were, respectively, 390 $cm^{-3}$, 94 $\mu m^2\ cm^{-3}$, and 6.1 $\mu m^3\ cm^{-3}$. The mean effective radius was 0.2 $\mu m$, greater than the value obtained in the previous analysis. The imaginary part of the refractive index was 0.01, and the SSA was 0.946 at 355 nm, 0.949 at 532 nm, and 0.942 at 1064 nm. The values were in agreement with those previously described, confirming the absence of black carbon. The volume size distribution reported in Figure 7 shows a less pronounced fine mode peak with concentrations significantly lower than those observed earlier (22:37–23:19 UTC). All microphysical properties are summarized in Table 2.

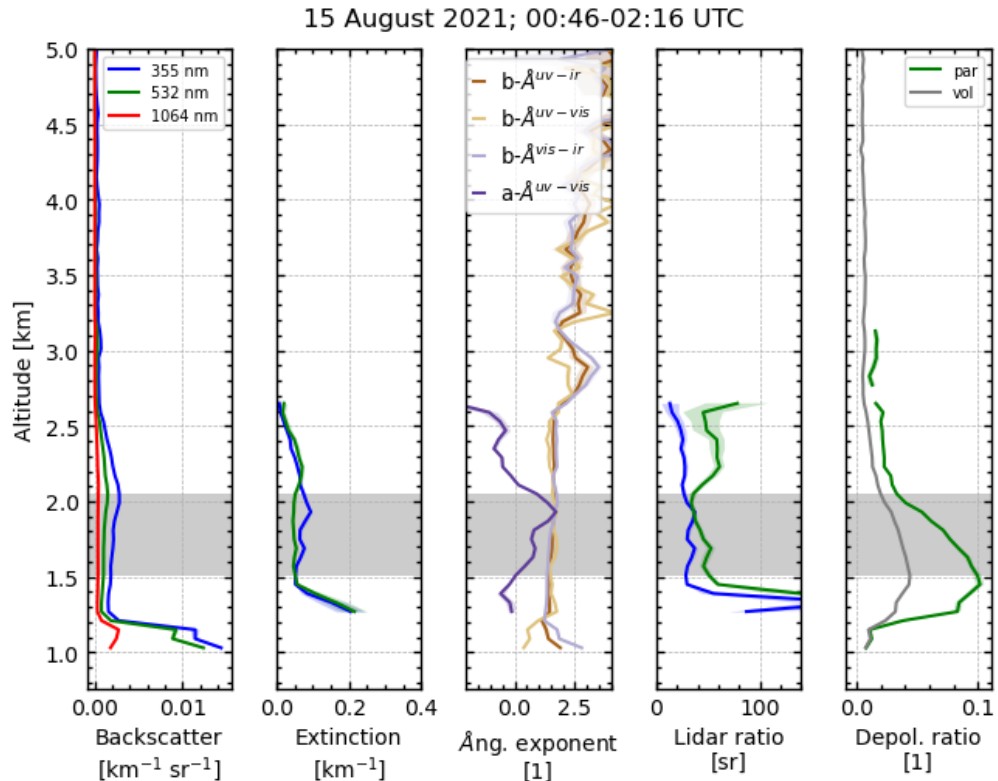

**Figure 6.** Vertical profiles of optical parameters were measured in Potenza on 15 August 2021 from 00:46 to 02:16 UTC with the multi-wavelength Raman lidar MUSA. The rectangle shows the range of altitude from 1510 to 2050 m asl used for obtaining the microphysical particle properties through the inversion method.

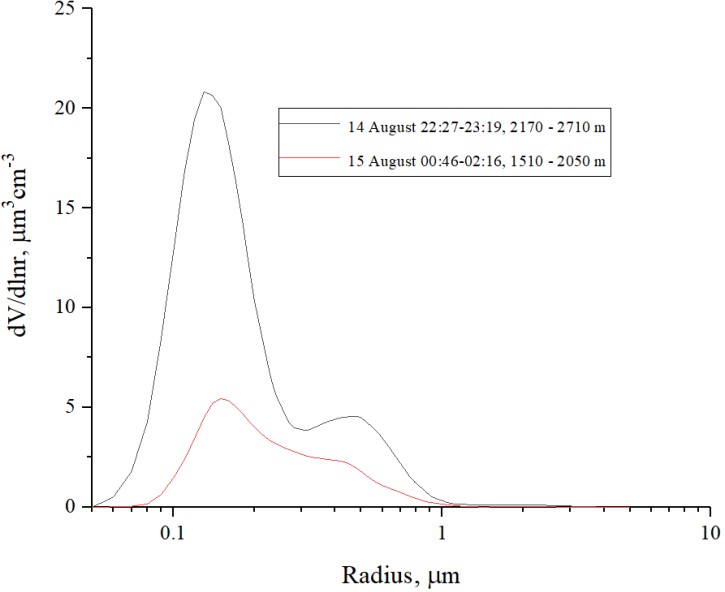

**Figure 7.** Volume size distribution retrieved starting from the lidar measurements of 14 August 2021 from 22:27 to 23:19 UTC and from 00:46 to 02:16 UTC.

**Table 1.** Values of LR at 355 and 532 nm, effective radius (Reff), real part (mR), and imaginary part of refractive index (mI) from different measurements of fresh biomass burning [12,38,43,48]. The time of travel for the smoke layer is estimated assuming a mean wind speed (i.e., 7.8 m/s) for the smoke layer as provided by the ECMWF forecast for the CIAO location.

| Date and time | LR355 | LR532 | Reff (µm) | mR | mI | Age |
|---|---|---|---|---|---|---|
| 15 August 2021, Potenza | 40 | 50 | 0.2 | 1.59 | 0.01 | <10 min |
| 14 August 2021, Potenza | 40 | 38 | 0.15 | 1.58 | 0.006 | <10 min |
| 17 October 2011, Evora [48] | 64 | 51 | 0.19 | 1.61 | 0.01 | 1 day |
| 9 August 2010, Bucharest [12] | 41 | 56 | 0.34 | 1.65 | 0.01 | 2 days |
| 29 July 2010, Bucharest [12] | 73 | 45 | 0.27 | 1.66 | 0.01 | 1 day |
| 22 July 2010, Bucharest [12] | 48 | 54 | 0.35 | 1.41 | 0.03 | 2 days |
| 15 August 2008, Silvicultura research site (Amazonia) [38] | 43 | 41 | 0.13 | missing | missing | 5 h |
| 26 September 2007, Granada [43] | 60 | 65 | 0.15 | 1.53 | 0.02 | 1 day |

**Table 2.** Values from 22:37 to 23:19 and 00:46 to 02–16 of effective radius (Reff), number concentration (N), surface concentration (S), volume concentration (V), real part (mR), and imaginary part of the refractive index (mI), and single scattering albedo (SSA) at 355, 532, and 1064.

| Date and Time | Reff (µm) | N (cm$^{-3}$) | S (µm$^2$ cm$^{-3}$) | V (µm$^3$ cm$^{-3}$) | mR | mR | SSA at 355, 532, 1064 nm |
|---|---|---|---|---|---|---|---|
| 15 August 2021, Potenza | 0.2 | 340 | 94 | 6.1 | 1.59 | 0.01 | 0.946, 0.949, 0.942 |
| 14 August 2021, Potenza | 0.15 | 2300 | 410 | 21 | 1.58 | 0.006 | 0.964, 0.967, 0.953 |

*3.2. Sun Photometer Measurements*

The co-located AERONET sun photometer was measured at 05:34 UTC on 15 August, three hours after the end of the lidar measurements. Figure 8 shows the wavelength dependence of the SSA from 440 to 1020 nm from 05:34 to 16:53 UTC on 15 August. The values agree with the retrieved values using the lidar measurements. At 440 nm, SSA values were around 0.95, in agreement with the values reported in the literature for smoldering forest fires observed with sun photometers [44]. Moreover, as already observed in similar studies [39], also here, single-scattering albedo decreased with the wavelength increase. It should also be noted that any differences were associated with the time disparity (>3 h, not active fire) and the difference in the measurement technique. AERONET provides columnar products, whereas the lidar-derived microphysical properties refer to the selected smoke layer.

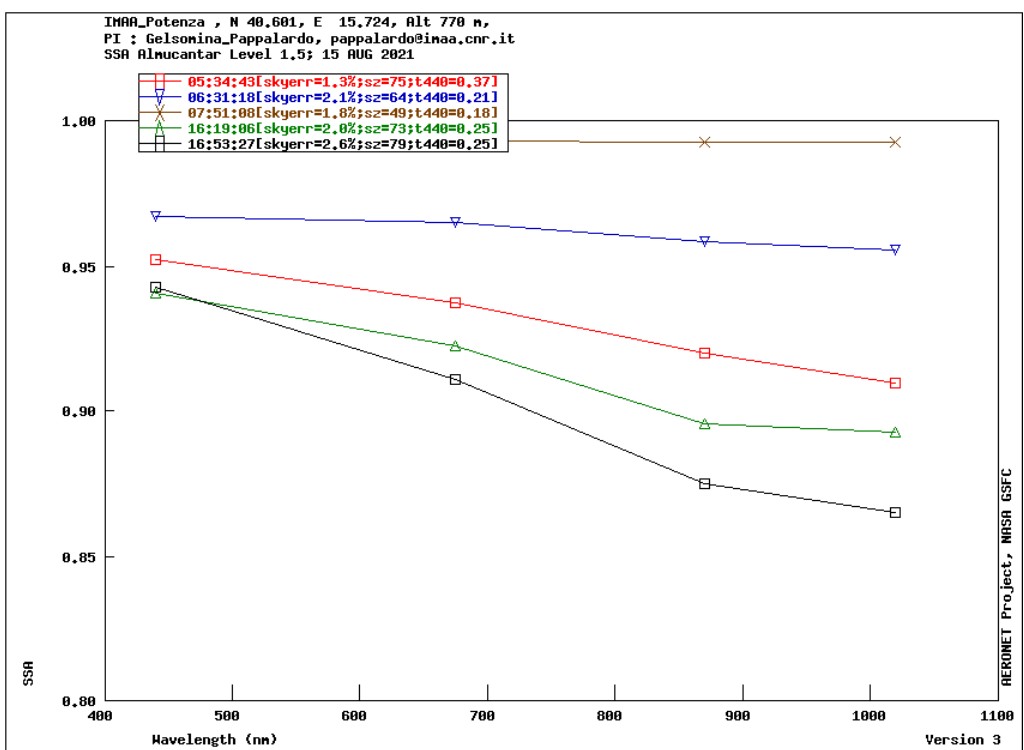

**Figure 8.** Spectral SSA provided by AERONET at 05:34, 06:31, 07:51, 16:19, and 16:53 UTC on 15 August 2021 at wavelength of 440, 675, 870, and 1020 nm.

## 4. Conclusions

This paper reports, for the first time, multiwavelength Raman lidar analysis of biomass burning aerosol formed a few minutes before the measurements. On 14 August 2021, an extremely fresh smoke plume was observed over the ACTRIS/EARLINET station of Potenza originating from a local forest only 1 km away. The fire started at 16:00 UTC and burned for about 7 h. The event was observed by the EARLINET/ACTRIS multiwavelength Raman lidar and a co-located AERONET photometer. This case study represents a unique event because of the very short distance between the fire source and the measuring station. The particles did not have the time to change during transport (smoke aging), and their optical properties only depended on vegetation type, combustion phase, and atmospheric conditions. Moreover, the aerosol was not contaminated by other particles from different sources. From 22:27 to 23:19 UTC, the measurements showed the presence of a thick smoke layer below 2.7 km. The results indicated fire particles with high scattering and surprisingly low absorption. The mean value of Ångström exponents was 1.5 at all wavelengths, and the lidar ratios were 40 sr at 355 nm and 38 sr at 532 nm, while particle depolarization was 0.025. Therefore, particles were spherical, moderately small, and weakly absorbing, probably due to a negligible contribution of black carbon. The inversion of three aerosol backscatter coefficients at 355, 532, and 1064 nm and two aerosol extinction coefficients at 355 and 532 nm were used to derive the microphysical properties of the particle inside the smoke layer. High values of surface and volume concentration and numeric density were retrieved: 410 $\mu m^{-2}$ $cm^{-3}$, 21 $\mu m^3$ $cm^{-3}$, and 2300 $cm^{-3}$, respectively. The size distribution showed a bimodal distribution in the accumulation mode with a peak at 0.13 $\mu m$. The medium value of the effective radius was 0.15 $\mu m$, indicative of small particles. The single scattering albedo values at 355, 532, and 1064 nm were approximately 0.96, the real part of the refractive index was 1.58, and the imaginary part was 0.006. These values were in agreement with the observation of aerosol characterized by a small absorption and a high scattering of the radiation. Similarly, for 00:46–02:16 UTC, the values of lidar ratios at 355 and 532 nm were 40 sr and 50 sr, respectively. The Ångström exponent at 355–532 nm was 1.5, while those at 355–1064 nm and 532–1064 nm were around 1.4. The values of

surface, volume concentration, and numeric density were 94 $\mu m^2\,cm^{-3}$, 6.1 $\mu m^3\,cm^{-3}$, and 390 $cm^{-3}$, respectively, with a mean effective radius of about 0.2 $\mu m$. The real part of the refractive index was 1.59, and the imaginary part was 0.01. The SSA values were 0.946 at 355 nm, 0.949 at 532 nm, and 0.942 at 1064 nm. Additionally, in this case, the predominance of the scattering over the absorption was considerable. AERONET measurements three hours after the end of the lidar observations confirmed the lidar observations despite the time discrepancy. The values of single scattering albedo of 0.95 at 440 nm were very similar to the corresponding values retrieved by lidar measurements. These results show that the compression of the climate impact of biomass burning aerosols is not entirely clear. Especially for small fires, the absorption component is severely underestimated. Further studies will be needed in the future. In particular, simultaneous measurements, obtained with lidar and chemical sensors, will improve the understanding of the radiative properties and the climatological impact of fresh biomass burning aerosols.

**Author Contributions:** : Conceptualization, B.D.R., L.M., A.A. and N.P.; methodology, I.V., L.M. and B.D.R.; validation, L.M., A.A. and G.D.; formal analysis, B.D.R., I.V., A.K. and S.S.; investigation, B.D.R., N.P., M.M. and D.S.; resources, B.D.R., L.M., N.P., A.A., A.K., S.S., A.G., A.F., G.D. and I.V.; data curation, C.D., F.A. and P.G.-C.; writing—original draft preparation, B.D.R.; writing—review and editing, L.M., A.A., N.P., B.D.R. and G.D.; visualization, B.D.R., A.F., A.K., N.P. and I.V.; supervision, L.M., N.P. and A.A.; project administration, L.M. and A.A.; funding acquisition, L.M. All authors have read and agreed to the published version of the manuscript.

**Funding:** The work was supported by the Sustainable Access to Atmospheric Research Facilities (ATMO ACCESS) and the Aerosol, Clouds, and Trace Gases Research Infrastructure Implementation (ACTRIS IMP) Projects. The work was also funded by CIR01_00015-PER-ACTRIS-IT, "Potenziamento della componente italiana della Infrastruttura di Ricerca Aerosol, Clouds, and Trace Gases Research Infrastructure-Rafforzamento del capitale umano", Avviso MUR D.D. no. 2595 del 24.12.2019 Piano Stralcio "Ricerca e Innovazione 2015–2017".

**Data Availability Statement:** EARLINET provides aerosol lidar profiles (https://www.earlinet.org/index.php?id=covid-19, accessed on 1 November 2020, https://www.earlinet.org/index.php?id=276, accessed on 26 September 2022). AREONET provides SSA data (https://aeronet.gsfc.nasa.gov/cgibin/data_display_inv_v3?site=IMAA_Potenza&nachal=0&year=2021&month=8&day=15&aero_water=0&level=2&if_day=0&if_err=0&place_code=10&DATA_TYPE=76&year_or_month=0, accessed on 26 September 2022).

**Acknowledgments:** We acknowledge ACTRIS for providing the CLU (2021) dataset in this study, which was produced by the Finnish Meteorological Institute, and is available for download from https://cloudnet.fmi.fi (accessed on 29 September 2022)/; and ECMWF for providing IFS model data (2020).

**Conflicts of Interest:** The authors declare no conflict of interest.

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
