# Peer review of "Characterization of Extremely Fresh Biomass Burning Aerosol by Means of Lidar Observations"

_remotesensing, doi:10.3390/rs14194984_

Round 1
Reviewer 1 Report
The paper shows some novel results for a small biomass burning event that is close to a lidar observatory. While the measurements occur towards the end of the event - the results are still scientifically interesting. My biggest issue with the paper is the vertical ranges chosen for the analysis. I think the results need to be redone for either a consistent altitude range for both cases or over the vertical range of the plume for each case. The current vertical ranges seem rather arbitrary.
Additional comments below:
Line 106 avoid colloquialism at beginning of sentence – use proper scientific writing
Fig 5 has very limited value – not sure it is required – the vertical scale is significantly different then the lidar plot – perhaps it is better to either state the value range in the text or provide a few profiles with context
Line 245 – over what vertical range are the values calculated – please specify
Line 266 – prefer author name to be used ahead of reference #
Line 283 – the plume is from 1700-2700 m yet most of the analysis is done between 2170 and 2710 m – even though there is “significant concentrations” of plume below 2170m – the range seems rather arbitrary and makes one wonder as to why the “full” range is not used – furthermore this vertical range is different than chosen a few hours later for the comparison – where again only a fraction of the vertical range of the plume is used – I think for comparison purposes one should either used the limits of the plume and redo the calculations or choose the same altitude range
Lines 273-282 – this section should be moved down – it interrupts the flow of the text where it currently is situated
Figs 9 & 10 are not of high enough quality for publication
Fig 8 – there are significant differences between Fig 8 and Fig 9 and 10 – there needs to be some discussion around this point
Reviewer 2 Report
Dear Authors,
Thank you for the interesting and meaningful manuscript. Using multi-wavelength lidar to study the microphysical properties of aerosol particles is very suitable. Also you provided a comparison of the co-located AERONET photometer data with time varies. The main innovation of this article is the description of the microphysical properties of fresh smoke, this is an enrichment of the aerosol historical database.
I still have some recommendations to improve its content.
1. The author believes that the generation time of fresh smoke is less than 10 min, but I have not found direct evidence. There may be some indirect proof, such as the forest is only 1 km away from the lidar location. The fire started at 16:00 UTC, while the earliest detection time of lidar is about 22:00 UTC. How to prove that the fresh smoke generation cycle is less than 10 min?
2. In figure 8, with the deposition of the aerosol and the boundary layer sinking, the concentration of particles with large particle size should rise at 00:46 to 02:16 UTC, while the detection results are the opposite, can you explain it?
3. Line 327, “02-16” should be corrected.
4. For the sun photometer, the error of the data in the early morning may be too large, especially at 5:34 and 6:31 in the morning. How is the data quality? If you want to draw the conclusion that "the concentration of the fine fraction decreases due to the coagulation processes", you may need to explain this issue.
5. Line 348, is there an extra enter?
Reviewer 3 Report
The paper entitled “Characterization of Extremely Fresh Biomass Burning Aerosol by Means of Lidar Observations” by De Rosa et al. presents the observation of freshly formed smoke particles from forest file by Raman lidar and sun photometer. The data is precious for the study of the evolution of newly emitted biomass burning particles. The manuscript is well-written and scientifically sound. I suggest publication of the high-quality paper after address the following question:
Microphysical inversion was made using the algorithms in the references. As I know, the algorithm makes an assumption that refractive indices of different wavelength (i.e. 355, 532, and 1064 nm) are the same. This would bring uncertainty when the proportion of organic carbon is large. However, that is quite difficult problem. I suggest that authors should discuss the problem when giving the result of microphysical inversion, especially the refractive index and the SSA.
Reviewer 4 Report
This article provides a unique study of very new biomass burning smoke prior to the modified changes in optical and microphysical properties occurring during transport.
The study uses the most cutting edge and mature retrieval algorithms combining Multiwavelength Raman Lidar and Constraining Multispectral Aerosol Optical Depth Measurements adopted within the Earlinet Network so accurate results can be expected.
The structure and the conclusions of the paper are straight forward and useful to the Air Quality community but unfortunately, the quality of the English writing is not as good as it shoiuld be. Many small clumsy english language constructs are used which shows a need for improved proof reading.
I have added the corrections I found in the abstract (as an attached PDF) as an example of the type of errors and their frequency and it is highly recommended that English reviewing services are used. Once corrected, I find the paper suitable for publication

Round 2
Reviewer 1 Report
Figure 8 seems to be missing and the text has been removed for Figure 9.
Author Response
Dear Reviewer,
thanks for the careful reading of the paper. We changed "Figure 9" into "Figure 8" in the caption. It was our mistake due to the fact that we removed one of the two figures. Therefore, Figure 9 does not exist anymore.